# The Effects of Vehicle Type, Transport Duration and Pre-Transport Feeding on the Welfare of Sheep Transported in Low Temperatures

**DOI:** 10.3390/ani11061659

**Published:** 2021-06-02

**Authors:** Francesca Carnovale, Jin Xiao, Binlin Shi, Tanel Kaart, David Arney, Clive J. C. Phillips

**Affiliations:** 1College of Animal Science, Inner Mongolia Agricultural University, 306 Zhaowuda Road, Inner Mongolia, Hohhot 010018, China; francesca.carnovale@student.emu.ee (F.C.); shibinlin@yeah.net (B.S.); 2Institute of Veterinary Medicine and Animal Sciences, Estonian University of Life Sciences, Kreutzwaldi 46, 51006 Tartu, Estonia; tanel.kaart@emu.ee (T.K.); david.arney@emu.ee (D.A.); 3Curtin University Sustainability Policy (CUSP) Institute, Curtin University, PKent St., Bentley, WA 6102, Australia; clive.phillips@curtin.edu.au

**Keywords:** sheep, animal welfare, transport, cold stress, cold wind, pre-feed, duration time

## Abstract

**Simple Summary:**

Sheep are often transported in cold conditions. Some of the coldest exist in northern China and we tested the effects of transporting sheep in these conditions. In order to find out how to limit any of the cold temperatures’ adverse effects, we tested journeys of two durations, in covered and open trucks, and investigated the effects of feeding sheep before transport. We found that sheep had less evidence of cold stress in a covered vehicle. Longer journeys actually allowed the sheep to warm up, but sheep appeared more stressed after the long journeys. Feeding the sheep before transport also helped to ameliorate the adverse effects of cold stress.

**Abstract:**

Low temperatures can provide a risk to the welfare of sheep during transport because of increased ventilation chilling the sheep, and we examined the importance of three factors—covering the vehicle, duration of transport, and feeding prior to transport—on the welfare of sixty transported 4-month-old Dorper × Mongolian female sheep in a cold climate. Sheep in a covered vehicle had greater increases in head and ear temperatures than those in an open vehicle. Sheep transported for 2 h increased their leg temperatures, whereas those transported for 1 h had reduced leg temperatures. Increases in non-esterified fatty acids (NEFA) and lactate dehydrogenase (LDH) in the blood samples during the long transport suggested that sheep had more muscular and metabolic activity, compromising their well-being. Feeding prior to transport did not affect body temperatures, but those not fed prior to transport had reduced alanine transferase, HSP and cortisol in their blood, whereas those that were fed had reduced NEFAs, LDH and creatine kinase. Prior feeding had no effect on the sheep temperature indices over a two-hour transport period. Thus, the sheep most at risk of the adverse effects of cold temperatures were those transported in open vehicles, those transported for a longer time, and those not fed before transport.

## 1. Introduction

Animals should be slaughtered as near to the point of production as possible. The journey time for slaughter animals should never exceed the physiological needs of the animal for food, water, or rest [1]. In China, more than 95% of the live animals slaughtered for the Hong Kong market are transported by truck and special cargo trains from inland provinces. This can include provinces in northeast China and Inner Mongolia, which involves travelling distances of over 3500 km [2].

The welfare of sheep during transport on the road can be influenced by many factors: the duration of the journey [3,4], stocking density [5], driving behaviour, vibration and noise, change in acceleration and cornering, road type [6,7], the design of the vehicle [1,8], loading and unloading [9,10], mixing with unfamiliar sheep or breeds, the novel environment, hunger, thirst and fatigue [1],temperature, humidity, excess ventilation and cold or heat stress [11,12,13].

Cold exposure is potentially one of the major stressors of livestock in winter. In a cold environment, homeothermic animals, including mammals, have to generate additional heat from their diet and body reserves [14]. In the fields, temperatures below the lower critical temperature and chilling winds can cause cold stress in livestock, as well as increasing their energy requirements, leading to decreases in productivity. For adult sheep, the lower critical temperature is –3 °C [2], below which daily metabolizable energy requirements of a 60 kg sheep increase by 0.14–0.64 MJ for each 1 °C decrease in ambient temperature from January to December in New Zealand [15]. A previous study has recommended that the appropriate ambient temperature for adult ewes in northwest pastoral regions of China should be above 2 °C [16]. Sheep are transported at temperatures far below this in the winter throughout the northern hemisphere, and in the region of this study, Inner Mongolia, winter temperatures can commonly be as low as −20 °C. The problems sheep may experience during transport at these temperatures, especially the effects of different journey lengths, vehicle design and level of feeding, have not been reported in the literature.

While sheep are considered more tolerant of transportation than other livestock, it can be stressful for both adult sheep and lambs, which show physiological responses such as increased heart rate, elevated plasma cortisol concentration and hyperthermia [17,18,19]. It has been noted that sheep do not experience changes in plasma osmolality even after 48 h without food and water, and after transport for 14 h, their immediate priority is for food; and only after 1–3 h do they start to drink [20]. It may therefore be more important to provide feed than water for sheep before their transportation.

Vehicle design is critical for the transport of animals [21]. It is recommended that transport vehicles should never be completely enclosed, as a lack of ventilation will cause excessive stress and even suffocation [21], but this could be different in cold temperatures. Additionally, it might also affect profitability—in pigs, cold stress can lead to weight loss [22,23] and mortality during transport [24]. In winter, the thermal properties of the trailer can be modified to reduce heat loss, including polystyrene insulation [25] and an insulated floor [26].

The duration of transport has also been investigated. Transport of sheep for 12, 30, and 48 h has resulted in reduced body weight and increased haemoconcentration on arrival [4]; however, after 9 h [27] or longer [28], sheep appear able to adapt to the stressors of the journey. However, this might not remain true in the context of very cold conditions.

The objectives of this research were to evaluate the effects of cold conditions on welfare during transportation of sheep and to establish what the perceived animal welfare issues in sheep transport are at cold temperatures under different transport conditions (an enclosed and open truck, with different durations of transport, and with or without feeding before transport). This will inform our understanding of the welfare issues associated with the transport of sheep in cold temperatures.

## 2. Materials and Methods

Sheep were cared for in accordance with the guidelines for animal experiments of Inner Mongolia Agricultural University. The experimental protocol (No: NND20212037) was approved by the Institutional Ethics Committee of Inner Mongolia Agricultural University.

### 2.1. Experimental Facilities

At the start and the endof the transportation, the location we tested was the Hailiutu experimental farm (Location: 40°51′~41°8′ North Latitude, 110°46′~112°10′ East Longitude) of the Inner Mongolia Agricultural University, Hohhot, Inner Mongolia, China. This sheep farm comprises an indoor barn area covered with straw and hay bedding and an outdoor area of yards, which, in winter, is covered with snow. Sheep were free to choose whether to be in the outdoor area or in the barn. Sixty 4-month-old Dorper × Mongolian female sheep, with average body weight of 18.02 ± 2.60 kg and fleece length of 40 mm, were randomly selected from a flock of eighty on the farm. They were divided into 20 groups of three similar sheep, each with its own identical house and outdoor area.

The sheep were fed in groups of three with the same ration: alfalfa hay (1.5 kg/head/d) first and then a pelleted feed (0.5 kg/head/d) one hour later, for thirty days. The diet contained 90.7% dry matter, 11.1 MJ/kg digestible energy and 13.6% crude protein. This was split into two feeds daily at 08:00/09:00 h and 15:00/16:00 h and was offered in the research facility. The physical composition of the diet included corn, soybeanmeal, cottonseedmeal, and a vitamin and mineral premix. Water was provided ad libitum both in the barn and in the outdoor area.

The transport journeys took place on 15 and 16 January 2020. Mean maximum and mean minimum temperatures on these two days were −13 °C/−21 °C and −11 °C/−18 °C; the humidity was 69% and 77%, respectively [29]. Three experiments were conducted with two treatments in each and 10 sheep in each treatment. The factors investigated in the three experiments were: Experiment 1—enclosing the vehicle (truck with or without a plastic cover and 1 h trip); Experiment 2—transport duration (1 or 2 h); and Experiment 3—pre-feeding (feeding the sheep before loading or not, 2 h trip). The 60 sheep were allocated to the six treatments so that the ten animals in each treatment had a similar body weight. Each sheep was used for one experiment only. Apart from Experiment 3, all sheep were held in an outdoor paddock and provided with access to alfalfa hay and fresh water for 2 h before each experiment.

#### Vehicles

All journeys were made using two similar vehicles: vehicle 1, a Beijing Futian truck manufactured in 2011 (Foton), and vehicle 2, a Wuling Rongguang truck manufactured in 2011. The two vehicles had the following respective dimensions: length 2.7 and 2.7 m, width 1.5 and 1.5 m and axle height 0.32 and 0.30 m. Both of the vehicle trays had 0.4 m solid walls, with a cage to height 1.1 m (truck 1) and 1.0 m (truck 2). In each experiment, trucks were allocated to treatments at random. The trucks were driven on a return route on a straight, single-lane carriageway (road numbers 038, 025 and 024) with minimal traffic at a consistent speed of 60 km/h. The route was chosen so that there was minimal traffic, no traffic lights and no stops, in order to minimise differences over time and between the journeys of the two vehicles.

### 2.2. Experimental Details

#### 2.2.1. Experiment 1. Enclosed vs. Open Truck

The first experiment compared sheep in a covered and an open truck, which both left the Hailiutu Experimental Farm at 09:00 h on 15 January. In the enclosed truck treatment, the rear tray of vehicle 1 was covered with polythene on the top and sides of the cage on all four sides, secured with rope to prevent the polythene moving during transport and also to prevent incursion of wind into the tray containing the sheep. The polythene was put in place immediately after loading. Truck 2 was used for the open vehicle treatment. The vehicles proceeded down the prescribed route for 1 h and then returned to the farm for a further 1 h.

#### 2.2.2. Experiment 2. Journey Duration

This experiment compared two journeys, one of 2 h and one of 1 h, made on 15 January on the same route as the first experiment, driven twice in the 2 h journey and once in the 1 h journey. The 2 h journey was from 14:00 h to 16:00 h and the 1 h journey from 14:30 h to 15:30 h, so the time midpoint was the same in both journeys. Both trucks were covered with plastic, as in Experiment 1′s covered vehicle. Vehicle 2 was used for the long duration journey and vehicle 1 was used for the short duration journey.

#### 2.2.3. Experiment 3 Feeding the Sheep before Transport

This experiment started at 11:00 h on 16 January. Sheep were fed 60 min before the journey commencement and at 20:00 h the night before, with 0.5 kg pellets /head and 1.5 kg alfalfa hay/sheep, offered in a trough. Sheep not fed before transport were fasted from 20:00 h on the night before, when orts were removed. Due to a breakdown of truck 2, journeys were made consecutively by vehicle 1. Sheep not fed before transport departed at 11:00 h and arrived back at 13:00 h; those fed departed at 14:00 h and arrived back at 16:00 h. In both groups, water was withheld for 12 h before transportation.

### 2.3. Measurements

#### 2.3.1. Environmental Parameters

Environmental temperature, humidity and wind speed were recorded by a local weather station at the farm at the time of the sheep transport [29]. In addition, during the transport, the wind speed was measured with an anemometer (Testo 416 Digital Mini Vane Anemometer, 99 Washington Street Melrose, MA 02176, USA) ten times, by holding the device outside the cab window.

#### 2.3.2. Physical Indices and Sheep Temperatures

Body weights of all sheep were measured to a precision of 0.1 kg 1 h before and after transportation. Starting body temperatures were recorded inside the building 1 h before transport, at the following locations on the sheep: head (midpoint between the ears), one ear (left), abdomen, and the foot/pastern region of the lower front leg, with the coronary band being the central point, using an infrared thermometer (Fluke MT4 max, Fluke Corporation P.O. Box 9090 Everett, WA 98206-90909090, USA).

After these initial measurements, the sheep were individually transferred outside next to the vehicle, for more detailed measurement of frostbite risk. A recording of the minimum temperature of one ear and the foot/pastern region was made to measure frostbite (Figure 1), using an infrared thermography camera (range −15–550 °C, UNI-T, UTi220A, Uni-trend Technology Co., Ltd., Songshan Lake National Hight-Tech Industrial, Guangdong Province, China), as these are the first areas to suffer from frostbite. After the journey, sheep temperatures were measured again at the same places in reverse order, first beside the vehicle and then in the building. For the outdoor recordings, as ambient temperatures were less than sheep body temperatures, it was necessary for the researcher to cup their ungloved hands around the back of the lower limb and ear to ensure that the recording of minimum temperature was of the sheep body part, not the environment.

#### 2.3.3. Physiological Indices

Two jugular blood samples were collected into heparinized tubes 1 h before and after the transportation for the analysis of stress indices. The blood samples were centrifuged and separated into plasma and serum components at the farm, which were then stored in a container with dry ice at a temperature of around −20 ℃, before analysis. Creatine kinase (CK), adrenocorticotropic hormone (ACTH), heat shock protein (HSP) alanine aminotransferase (ALT), lactic acid dehydrogenase (LDH), free fatty acid levels (NEFA), catecholamine (CA) and cortisol concentration were determined with commercial ELISA kits (Nanjing Jiancheng Institute of Bioengineering, Nanjing, China and Ruixin Biological Technology Co., Ltd. Quanzhou, China), according to the manufacturer’s instructions. Glucose was measured using a HITACHI 7020 automatic biochemistry analyser.

### 2.4. Statistical Analysis

Experiments were individually analysed for statistical differences between treatments. Based on temperatures and blood parameters measured before departure and after return, the changes during the transport were calculated for each animal. These changes were compared between treatments using a general linear model, including treatment as a fixed effect and value at departure as a covariate. In an attempt to find the common patterns in measured temperatures and blood parameters, and to study the effect of transportation under different conditions on those patterns, a principal component analysis was conducted for each experimental dataset. Results were considered statistically significant at *p* ≤ 0.05. Statistical analyses were performed, and figures were constructed using R version 3.5.3 (R Foundation for Statistical Computing, Vienna, Austria).

## 3. Results

### 3.1. Experiment 1

The ambient conditions recorded at 09:00 and 11:00 h, the start and end of the journeys, respectively, were as follows: environmental temperature −17 and −13 °C, 65 and 56% humidity, and 6 and 6 km/h wind speed [29]. The average wind speed recorded from the truck was 17.0 (±0.86 SEM) m/s.

#### 3.1.1. Indoor Average Temperatures

All body temperatures increased during the journey, particularly the head and, to a lesser extent, the ear temperature (Table 1). The increases in head and ear temperatures in the enclosed vehicle were greater than those in the open vehicle. There was no difference between the enclosed and open vehicles in the temperature increases of the abdomen and leg. Mean start and end temperatures and individual differences are presented in Appendix A.

#### 3.1.2. Outdoor Minimum Temperatures

Both ear and pastern minimum temperatures increased during the journey, particularly the ear. There were no differences between covered and uncovered vehicles in the minimum temperature of the ear or pastern.

#### 3.1.3. Blood Parameters

Heat shock proteins declined in both groups, but the reduction was much more pronounced for sheep in the enclosed vehicle (Table 1, Appendix A). Changes in the other blood parameters were not different between treatment groups.

#### 3.1.4. Principal Components

A principal component analysis confirmed that the first principal component related mainly to biochemical parameters, in particular ACTH, LDH, NEFA, CA and CK on the one side and glucose on the other. The second principal component showed that the changes in ear and head temperatures were antagonistic to changes in HSP. The graph (Figure 1 and Figure 2) of individual sheep changes against the first two principal components demonstrates that the sheep in the enclosed vehicle changed most in their ear and head temperatures, in comparison with the sheep in the open vehicle.

### 3.2. Experiment 2

This experiment was conducted at higher temperatures than in experiment 1. The ambient conditions recorded at 14:00 and 16:00 h, the start and end of the long journey, respectively, were as follows: environmental temperature −11 and −8 °C, 67 and 53% humidity, and 6 and 2 km/h wind speed [29]. The average wind speed recorded in the truck was 13.3 (±0.49 SEM) m/s.

#### 3.2.1. Indoor Average Temperatures

All body part temperatures increased during the journey, except leg temperature in the sheep transported for the short duration, in which case there was a small reduction. The increases in head, ear and abdominal temperature did not differ for the two durations of transport, but the leg temperatures increased for sheep in the long duration transport;in contrast, as noted above, those in the short duration transport reduced leg temperature (Table 2, Appendix A).

#### 3.2.2. Outdoor Minimum Temperatures

The minimum temperatures of the sheep’s ears tended to increase when they were transported for two hours but decreased when they were transported for one hour (*p* = 0.052). The maximum ear temperature was not affected by treatment. However, the minimum and maximum pastern temperatures decreased less (minimum) or increased in the 2 h transport compared with a decrease for 1 h transport.

#### 3.2.3. Blood Parameters

The blood analysis showed that ALT, cortisol and CK had little change over the short duration transport but increased significantly in the long duration transport. ACTH tended to decrease in the short duration transport and increase in the long duration transport (*p* = 0.08). NEFA and LDH declined in the short duration transport and increased in the long duration transport (Table 2, Appendix A).

#### 3.2.4. Principal Components

The principal component analysis indicated similar first and second components to Experiment 1 (Figure 3). The long duration transport indicated more of a focus on biochemical responses at the end of the journey, in comparison to the short duration, which demonstrated more leg temperature responses.

### 3.3. Experiment 3

At the start and return of the first journey and the start and return of the second journey, the ambient conditions were −13, −10,−9 and −8 °C; 72, 67, 62 and 58% humidity; and 2, 2, 4 and 6 km/h wind speed, respectively [30]. The average wind speed recorded in the truck was 18.94 (±0.25 SEM) m/s.

#### 3.3.1. Indoor Average Temperatures and Outdoor Minimum Temperatures

All body part temperatures increased during the journey, except leg temperature in the sheep transported after being fed, in which case there was a small reduction (Table 3, Appendix A). Feeding prior to transport had no effect on the sheep temperature indices recorded inside and on the minimum temperature of the ear or pastern recorded outside before and after the two-hour transport period.

#### 3.3.2. Blood Parameters

ALT increased in sheep not fed and declined in those fed before transport (*p* < 0.001), HSP increased more when there was no feeding, cortisol declined in fed sheep but showed little change in sheep not fed, NEFAs and LDH declined considerably in fed sheep but increased in sheep not fed, and there was a tendency for the same pattern in CK (Table 3, Appendix A).

#### 3.3.3. Principal Components

Principal components were similar to Experiments 1 and 2, except that HSP followed a similar pattern to head temperature and the sheep that were not fed tended to have more response in these parameters after the journey (Figure 4).

## 4. Discussion

### 4.1. The Effect of Enclosing the Truck

It is important to consider whether the temperatures experienced by the sheep in this study were below their lower critical temperature. The environmental temperatures recorded during Experiment 1 were the lowest of any in this study (before the transport, −17 °C). The ranges in temperature that a fully fleeced adult sheep can cope with, according to Taylor (1992) [30], are between −12 °C and 32 °C. To cope with the low temperatures, sheep increase the production of metabolic heat [31,32], but this ameliorative strategy has its limitations. Neither temperate nor humidity were not recorded from within the truck, which may be considered a limitation to the findings. However, these might be expected to vary within the truck and the animal-sourced measures of temperature differences that were recorded were the focus of this study and reflect the experience of the individual animals.

The lower critical temperature that sheep can cope with depends on the wool length [12,33]; For sheep with 50-mm-thick wool, it is approximately −5 °C and for sheep with 70-mm-thick wool, −18 °C. In this experiment, fleece thickness was less than these, at 40 mm, indicating that it is likely we were operating below the lower critical temperature for the sheep in all three experiments. These thresholds also depend on other weather conditions, such as wind chill and humidity [12]. Dabiri et al. (1995) [34] considered that sheep become cold stressed at an ambient temperature of +7 °C if the weather is windy and wet. However, according to a recent farm study, sheep preferred to be outside during winter in a cool and dry climate, at temperatures as low as −20 °C [35]. The ‘wind chill’ factor can double heat loss. A wind speed of 17 m/s at −13 °C has a wind chill outcome of −38 °C [36]. If the cold persists, the body temperatures of sheep drop, ultimately causing hypothermia and frostbite [37]. Hypothermia symptoms are the mucous membranes turning pale to white and cold legs [37].

Young and recently shorn sheep are particularly susceptible to frostbite and loss of body heat during transport [38], and the sheep in this trial were approximately four months of age. Yet, we recorded an increase in average leg temperatures at the end of the transports in both vehicle designs (Appendix A), which could have been the result of sheep huddling together during the voyage. During transport, the sheep tend to form a group and retain warmth by huddling [39], thereby avoiding the wind [40,41]. The colder temperatures of the head and ears recorded in the open than enclosed vehicle (*p* < 0.01) (Table 1) demonstrate that shelter against the cold wind reduces the possibility of cold stress during transport [36]. Thermal differences between the sheep in the different trucks were not found in the legs and abdomen, likely due to the presence of sides in both open and closed trucks, which did not let the wind pass into direct contact with the lower body parts of the sheep.

The infrared temperature data were not different between the two vehicle designs. There was therefore no evidence of higher incidence of frostbite in the sheep in the open trucks, although this might have been due to the short period of exposure.

Of all the blood parameters recorded, only heat shock proteins (HSP) were significantly different between the two treatments. HSP are synthesized in response to different stressors [42], especially under conditions of temperature stress [43,44]. HSPs decreased more in the enclosed vehicle than in the open vehicle, where thermic stress through cold of the sheep might be expected due to less exposure to wind and cold temperatures from outside. It has been shown that the expression of HSP can occur after cold temperature change, such as in fish and rats [45,46]. In sheep, however, there is no evidence of HSP expression in cold stress, but as shown from the PCA (Figure 1 and Figure 2), HSP concentration was related in a vertical pattern to all of the temperature variables, and also both glucose and cortisol. Cortisol is a stress hormone [17], and the raised glucose might be indicative of increased metabolism and/or muscle activity.

### 4.2. Effect of Duration of Transport in Cold Conditions on Sheep Welfare

There was an increase in the temperature of all body parts for both journey durations. However, the leg temperatures increased more during the 2 h journey compared to the 1hjourney, which may have been due to the sheep spending more time huddling together and a longer time in contact with the warm floor heated by the truck engine’s exhaust pipes that ran underneath.

Cortisol is an important indicator of stress and is commonly used in sheep to assess the effects of transport and its duration [17,47]. For the two-hour transportation compared with the one-hour transportation, cortisol increased more, by 9% compared to 4% (Table 2). Fordham et al. (1989) [48], Broom et al. (1996) [9] and Androine et al. (2008) [49] also found that cortisol concentrations increased in the plasma of sheep during the first hours of transport.

Sheep respond during transportation to both adrenocortical and muscular stress [50]. Creatine kinase, the enzyme of choice in sheep studies because of its sensitivity and rapid serum increase after muscular damage [51], increased significantly in the longer duration transport, and previous studies in sheep have also found this for transported sheep [52,53]. Our trial used four-month-old ewes whereas Zhong et al. (2011) [52] found that adult sheep are more susceptible to muscular damage during transport than younger animals. Muscle damage could be due to movement of the animals, bumping into each other or the sides, with more risk of this during a longer period of transport, especially if fatigue sets in [54]. This would be affected by driving style: Cockram et al. (2004) [6] reported muscle damage due to driving events, which caused around 80% of the losses of balance of sheep during transport.

Alanine amino transferase (ALT) also increased significantly over the longer transport. This is an enzyme of the liver, and its upsurge in the blood follows modifications of normal liver function [55]. Fluctuations in the values of ALT in animals may indicate intensification of metabolic processes or metabolic disorders [56]. High concentrations of ALT could be an index of activity in the blood for the metabolic processes involved in carbohydrate, protein, and fat conversion [56,57]. The rates of metabolism increase during stressful conditions [58] and also when skeletal muscle is regularly contracting [59].

Lactate dehydrogenase (LDH) declined in the short transport but increased in the longer transport (*p* < 0.01). This indicates intense muscular activity and muscular damage [60,61], which might indicate more shivering in these sheep. NEFAs also rose significantly in the longer transport. The blood concentration of NEFA is related to adrenaline and glucagon. Adrenaline and glucagon activate lipase, inducing hydrolysis of fatty acids reserves and therefore, the production of NEFA and ketone bodies [62]. As the journey progressed longer, imbalances in the metabolic activities of the sheep consequent to stress also rose. These parameters all suggest that the sheep had entered a period of stress by the end of the longer period of transport.

Changes in the blood parameters were also evident in the PCA analysis (Figure 3) where the temperature and the duration of the transport were related to the changes in the blood parameters.

Infrared thermography is a practical, non-invasive technique to determine changes in the body temperature of animals [63]. The imaging IRT (minima) decreased in the temperature of the ears in the short transport but were largely unchanged after the 2 h transport. The feet temperatures were also significantly different, declining more in the short than the longer transport period. The comparatively warmer temperatures recorded by the IRT during the longer voyage could indicate fear, pain, and stress, as indicated by changes of at least 0.4 °C in the eyes and 0.9 °C in the limbs [64,65]. However, the determination of IRT could have been affected by error factors such as ambient temperature, humidity, wind speed, camera positioning, and animal factors, such as skin and hair colour, unexpected movement, health status, and time of feeding [66,67,68]. The trucks were also enclosed, so that these sheep were not exposed to wind chill, unlike the area they were kept prior to loading, or during loading. After two hours, these responses may indicate that the longer journey allowed the sheep to cope with the cold temperatures. However, for journeys of longer duration, the capacity of the sheep to cope in this way might well become exhausted.

### 4.3. Effect of Pre-Transport Feeding on Sheep Welfare during Transport in Cold Conditions

There was a small reduction in the leg temperatures in the sheep transported after being fed, possibly because blood flow was concentrated around the gastrointestinal tract and not in the measured body parts of those sheep [69]. All other body part temperatures and IRT temperatures increased.

Concerning blood parameters, ALT decreased in the fed animals and increased in the sheep that were not fed prior to transport. ALT is involved in gluconeogenesis [70,71] and the treatment difference could be explained by the need to synthesise glucose in the fasted sheep.

As with ALT, HSP, cortisol, NEFA, LDH, and CA significantly (and CK with a tendency) increased in the sheep that were not fed prior to transport. This indicates that the sheep that were not fed before transport experienced greater stress during the transport than those that were fed. ACTH declined in both groups, but was not different between treatment groups, which might be expected as this is produced in response to short-term stress [72]. However, we noted from the PCA results (Figure 4) that ACTH, as well as glucose, HSP, CK and LDH, had a strong relationship with temperature parameters.

## 5. Conclusions

During transport in cold conditions, using a vehicle that is completely enclosed is important to reduce the cold stress suffered by sheep, even over a short 1 h journey. A longer (2 h) journey, despite allowing the sheep to warm up, resulted in an increase in a range of parameters indicating stress. These also included an increase in indicators, suggesting the mobilisation of body reserves, which is common in sheep for stabilising body temperature in cold conditions. Over a journey of 2 h duration, this probably allowed the sheep to cope, but longer journeys may present a problem. Feeding sheep prior to transport can assist in ameliorating the stressors experienced by the sheep during transport in cold conditions.

## Figures and Tables

**Figure 1 animals-11-01659-f001:**
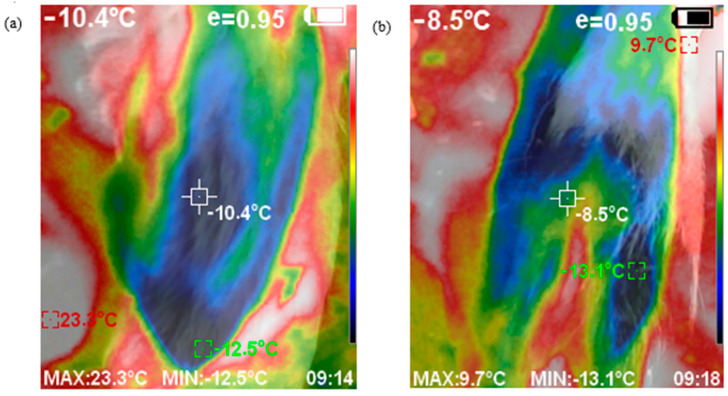
Example ofrainbow colour palette of front-on view thermograms of a sheep’s ear (**a**) and foot (**b**). Measured variables, outlined by the square, where the red is maximum temperature, green minimum temperature and white is the centre point of the camera IRT. The coloured index depicted temperatures from hot to cold using the colours white, red, yellow, green and blue.

**Figure 2 animals-11-01659-f002:**
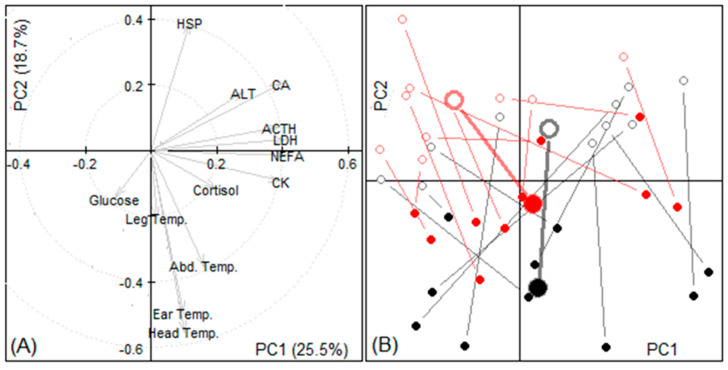
Principal components identified as PC1 and 2 in experiment 1. (**A**) The weights (eigenvectors) of variables in the first two principal components (PC). (**B**) Location of individual sheep samples according to their first two PC scores sorted by treatment group (black: enclosed vehicle, red: open vehicle,) and time (departure: empty circles, arrival: filled circles); samples from the same animal are joined by a line, larger circles indicate treatment groups’ means.

**Figure 3 animals-11-01659-f003:**
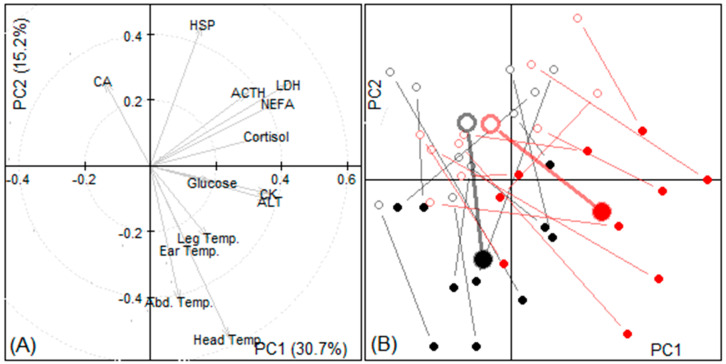
Results of principal component analysis of experiment 2 data. (**A**) The weights (eigenvectors) of variables in the first two principal components (PC). (**B**) Location of samples according to their first two PC scores sorted by treatment group (red: duration of transport 2 h, black: duration of transport 1 h) and time (departure: empty circles, arrival: filled circles); samples from the same animal are joined with a line, larger circles indicate groups’ means.

**Figure 4 animals-11-01659-f004:**
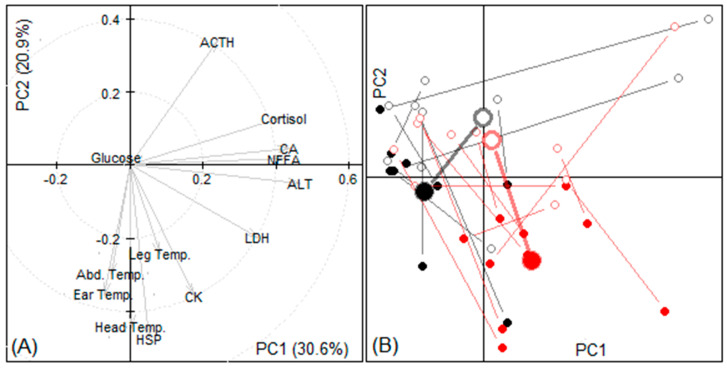
Results of principal component analysis of experiment 3 data. (**A**) The weights (eigenvectors) of variables in the first two principal components (PC). (**B**) Location of samples according to their first two PC scores sorted by group (red: without pre-feeding, black: with pre-feeding) and time (departure: empty circles, arrival: filled circles); samples from the same animal are joined with a line, larger circles indicate groups’ means.

**Table 1 animals-11-01659-t001:** Changes in temperatures, infrared temperature images and blood parameters during transport (values at arrival minus values at departure) of sheep (n = 2 × 10) in Experiment 1. Mean changes are adjusted for departure values and are presented with 95% confidence intervals. *p*-values indicate the statistical significance of treatment effects on the adjusted change.

Variable	Change During Transport		*p*-Value
	Enclosed Vehicle	Open Vehicle	
Indoor temperature, °C			
Head	+16.4 (14.8, 18.0)	+11.8 (10.3, 13.4)	<0.01
Ear	+9.9 (8.0, 11.7)	+2.0 (0.1, 3.8)	<0.01
Abdomen	+1.2 (0.3, 2.2)	+1.3 (0.3, 2.2)	0.89
Foot/pastern	+0.6 (−2.4, 3.6)	+0.2 (−2.8, 3.2)	0.84
Outdoor minimum temperature, °C			
Ear	+9.7 (7.7;11.6)	+8.5 (6.5;10.4)	0.39
Foot/pastern	+5.8 (4.6;7.1)	+6.1 (4.7;7.3)	0.85
Blood parameters			
ACTH, ng/L	+2.7 (−7.9, 13.3)	+0.2 (−10.3, 10.8)	0.75
ALT, U/L	−1.35 (−3.49, 0.79)	−0.15 (−2.29, 1.99)	0.41
HSP, ng/L	−303.9 (−495.7, −112.2)	−24.8 (−216.5, 166.9)	0.04
Cortisol, μg/L	+2.26 (−1.07, 5.58)	+2.96 (−0.37, 6.29)	0.76
NEFA, mmol/L	+7.0 (−10.1, 24.1)	+15.2 (−1.9, 32.3)	0.51
LDH, IU/L	−0.7 (−16.3, 14.8)	+7.1 (−8.4, 22.6)	0.46
CA, ng/L	−5.0 (−13.2, 3.2)	−0.8 (−9.0, 7.4)	0.46
CK, IU/L	+1.67 (−0.29, 3.63)	−0.48 (−2.44, 1.48)	0.14
Glucose, mmol/L	−0.01 (−0.83, 0.82)	+0.87 (0.04, 1.70)	0.13

ACTH = adrenocorticotropic hormone; ALT = alanine aminotransferase; HSP = heat shock protein; NEFA= free fatty acid levels; LDH = lactic acid dehydrogenase; CA = catecholamine; CK = creatine kinase.

**Table 2 animals-11-01659-t002:** Changes in temperatures, infrared temperature images and blood parameters during transport (values at arrival minus values at departure) of sheep (n = 2 × 10) in experiment 2. Mean changes are adjusted for departure values and are presented with 95% confidence intervals. *p*-values indicate the statistical significance of the treatment group effect on the adjusted change.

Variable	Change During Transport		*p*-Value
	Duration of Transport 1 h	Duration of Transport 2 h	
Indoor temperature, °C			
Head	+10.0 (7.4, 12.6)	+9.4 (6.8, 12.0)	0.73
Ear	+3.3 (0.0, 6.7)	+1.3 (−2.0, 4.7)	0.45
Abdomen	+1.4 (0.7, 2.1)	+0.5 (−0.3, 1.2)	0.08
Foot/pastern	−0.5 (−1.7, 0.7)	+1.9 (0.7, 3.0)	0.01
Outdoor minimum temperature, °C			
Ear	−2.2 (−4.3;−0.1)	+0.8 (−1.2;2.9)	0.05
Foot/pastern	−4.6 (−6.5;−2.8)	−1.1 (−2.9;0.7)	<0.01
Blood parameters			
ACTH, ng/L	−5.8 (−18.8, 7.1)	+10.7 (−2.2, 23.7)	0.07
ALT, U/L	+0.8 (−1.19, 2.86)	+6.5 (4.43, 8.48)	<0.01
HSP, ng/L	−127.8 (−238.6, −17.1)	−126.6 (−237.3, −15.8)	0.98
Cortisol, μg/L	+0.1 (−2.25, 2.46)	+5.2 (2.87, 7.58)	<0.01
NEFA, mmol/L	−19.7 (−40.0, 0.6)	+21.4 (1.1, 41.7)	<0.01
LDH, IU/L	−18.5 (−43.2, 6.2)	+38.4 (13.8, 63.1)	<0.01
CA, ng/L	−6.4 (−24.0, 11.3)	−14.4 (−32.0, 3.3)	0.55
CK, IU/L	+0.6 (−1.84, 3.02)	+5.3 (2.85, 7.71)	0.01
Glucose, mmol/L	+0.03 (−1.03, 1.08)	+0.74 (−0.31, 1.80)	0.33

ACTH = adrenocorticotropic hormone; ALT = alanine aminotransferase; HSP = heat shock protein; NEFA= free fatty acid levels; LDH = lactic acid dehydrogenase; CA = catecholamine; CK = Creatine kinase.

**Table 3 animals-11-01659-t003:** Changes in temperatures, infrared temperature images and blood parameters during transport (values at arrival minus values at departure) of sheep (n = 2 × 10) in experiment 3. Mean changes were adjusted for departure values and are presented with 95% confidence intervals. *p*-values indicate the statistical significance of treatment group effects on the adjusted change.

Variable Pre-Transport Feeding	Change During Transport		*p*-Value
	Yes	No	
Indoor temperature, °C			
Head	+10.1 (6.8, 13.4)	+11.8 (8.5, 15.0)	0.49
Ear	+2.9 (−0.8, 6.6)	+6.6 (2.9, 10.3)	0.16
Abdomen	+0.7 (0.0, 1.3)	+1.4 (0.8, 2.1)	0.13
Foot/pastern	−0.2 (−2.1, 1.7)	+2.0 (0.1, 4.0)	0.10
Outdoor minimum temperature, °C			
Ear	+5.2 (1.8;8.6)	+5.5 (2.09;8.9)	0.90
Foot/pastern	+6.1 (4.4;7.8)	+4.7 (2.9;6.4)	0.24
Blood parameters			
ACTH, ng/L	−4.8 (−11.9, 2.3)	−8.6 (−15.7, −1.5)	0.43
ALT, U/L	−2.9 (−5, −0.93)	+4.6 (2.52, 6.59)	<0.01
HSP, ng/L	+90.3 (−57.4, 238.1)	+309.7 (162.0, 457.5)	0.04
Cortisol, μg/L	−3.9 (−6.02, −1.88)	−0.2 (−2.25, 1.89)	0.01
NEFA, mmol/L	−20.1 (−27.3, −12.8)	+18.0 (10.8, 25.3)	<0.01
LDH, IU/L	−18.5 (−48.1, 11.0)	+26.6 (−2.9, 56.1)	0.03
CA, ng/L	−8.5 (−18.1, 1.1)	+8.9 (−0.8, 18.5)	0.01
CK, IU/L	−1.7 (−5.81, 2.42)	+3.7 (−0.41, 7.82)	0.06
Glucose, mmol/L	+0.1 (−0.55, 0.63)	+0.2 (−0.37, 0.81)	0.66

ACTH = adrenocorticotropic hormone; ALT = alanine aminotransferase; HSP = heat shock protein; NEFA= free fatty acid levels; LDH = lactic acid dehydrogenase; CA = catecholamine; CK = Creatine kinase.

## Data Availability

The raw data have not been published or stored elsewhere, but are available on request from F.C.

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
