# Peer review of "The Effects of Vehicle Type, Transport Duration and Pre-Transport Feeding on the Welfare of Sheep Transported in Low Temperatures"

_animals, 2021, doi:10.3390/ani11061659_

Round 1
Reviewer 1 Report
Overall, it is an interesting study showing the effects of cold weather journey on adult sheep. The variables measured allow us to infer the impact of travel conditions on body temperature and biochemical stress response. However, there is no evidence of measurement of the temperature-humidity regime during journey. Recent studies indicate that temperature-humidity patterns inside the truck during travel often differ from outside temperature-humidity. In cold weather, animals will tend to produce heat and maintain their body temperature within their thermo-neutral range. After loading, and at high stocking densities, the humidity and temperature within the truck rise quickly, creating a microclimate which favors dehydration. Therefore, the lack of microclimate monitoring during the trip is a limitation of this study and should be discussed by the authors. There is also a need for the authors to discuss more about stress hypothermia; there are some articles on sheep on long and short journeys that have measured this parameter with other techniques. I think the discussion should include the appropriateness of the use of the infrared thermometer and thermographic camera used in this study versus portable devices used in similar studies.
Other comments
L23 Indicate the total number of transported animals included in the study, as well as sex, breed or genotype.
L63 It also causes marked hyperthermia regardless of the length of the trip (see Pascual Alonso 2017 in Journal of Animal Physiology and Animal Nutrition and Miranda et al., 2018 Tropical Animal Health and Production).
L99 randomly?
L111 It is necessary to clarify that it was a one-hour trip.
L155-159 Temperature-humidity monitoring inside the truck to record micro-environmental changes during the trip?
L180 “analysis of physiological stress response”
Reviewer 2 Report
Dear author, this is an interesting field of research and it provides good hints for practitioners. But some details should be described closer and the conclusions don’t thoroughly fit to the research. Moderate spelling changes are required.
In detail:
Line13: delete the word “on”
Line 25: Please mention that NEFA and LDH were measured in blood. I don’t understand this sentence; I think it is not finished?
Line 30: delete “those” after the word “transported”
Line 40: delete the bracket after “km”
Line 54: delete “the” before “New Zealand”
Line 56: Could you please mention which sheep breed(s) you are talking about?
Line 60: by whom are these aspects not understood? By the practitioners? By scientists?
Line 80: is it really the effects of cold conditions on welfare management you want to look at? In your study you show the effects of cold conditions and of some management conditions on animal welfare aspects. But you don’t show the effects on management! Please change that.
Line 98: delete “for the study”. Why did you wait for two days to select them randomly? Please explain.
Lines 104-105: Could you give a closer description of the feed? (at least dry matter, MJ (energy) and protein contents)
Lines 107-108: Mean, maximum and minimum temperatures of two days: this should be 2 x 3 figures: But here you show only 2 x 2 figures. Please complete
Line 147: replace “on the previous night” by “the night before”
Line 149: replace “on the previous night” by “the night before”
Line 149: I don’t understand the word “orts”
Line 168-169: replace “to beside the vehicle” by “next to the vehicle”
Lines 170-171: please explain closer how you detected frostbite; which were the indications for frostbite?
Line 173: replace “beside the vehicle” by “next to the vehicle”
Line 182: add the word “of” after temperature
Line 204: delete “and 6”
Lines 230, 276, 306: in the table descriptions you write n=10, but on line 110 you wrote that you used 10 sheep for each treatment; so in the tables we see the results of 20 sheep = 2x 10 = 10 for each treatment?!
Line 242: replace “greed” by “green”
Lines 241-244: This is more or less a methods description. But what is the interpretation of what you can see here? Can you detect frost bites? Or where / how could you detect them? Or is it just an example? For what?
Line 333: You say that sheep preferred to be outside in winter in a cool climate. But I think this is only true if it is a cool and dry climate.
Line 334: Is that really true: at 13°C it has a wind chill outcome of -38°C? I think this can’t be true. Maybe the first figure should be -13°C?
Line 428: delete “to” after “by”
In the discussion part you should mention that the journeys in your experiments were much shorter than the ones you mentioned in the introduction (up to 3’500 km), and that you don’t know what would happen if they would have been much longer.
You should also try to explain why
Line 438: It cannot be a conclusion of your study that a completely enclosed vehicle is important to reduce cold stress, because you have not done this research. On line 132 you mentioned that the vehicle you used in the study was only covered on top and had uncovered sides. So, you can’t say how it would have been with a completely enclosed vehicle!
Line 444: That “longer journeys may present a problem” is not easy to know and not a result of your study. You can say this in the discussion part and ideally underline it with a reference. But it is not appropriate and not consistent to call this a conclusion of your study, because you even explain on lines 368-369 that the sheep were a longer time in contact with the warm floor during the longer journey in experiment 2. This could also be true for even longer journeys…
Please rewrite your conclusions part.
Reviewer 3 Report
Sheep are considered to be relatively resistant to transport stress including temperature changes. There were not as many research studies published as for other livestock species, though. Thus, this paper is a valuable contribution to the knowledge in this area providing data on how sheep cope with cold temperature during transport and what measures can effectively improve their welfare when transported in cold regions.
Author Response
Thank you for your comments